# Brief communication: Application of a muonic cosmic ray snow gauge to monitor the snow water equivalent on alpine glaciers

Rebecca Gugerli[1,*,**], Darin Desilets[2], and Nadine Salzmann[1,***,****]

[1]Department of Geosciences, University of Fribourg, Switzerland
[*]now at Environmental Remote Sensing Laboratory, École Polytechnique Fédérale de Lausanne (EPFL), Lausanne, Switzerland
[**]now at Radar, Satellite and Nowcasting Division, MeteoSwiss, Locarno-Monti, Switzerland
[2]Hydroinnova LLC, Albuquerque, USA
[***]now at WSL Institute for Snow and Avalanche Research SLF, Davos, Switzerland
[****]now at Climate Change, Extremes, and Natural Hazards in Alpine Regions Research Center CERC, Davos, Switzerland

**Correspondence:** Rebecca Gugerli (rebecca.gugerli@epfl.ch)

**Abstract.** Monitoring the snow water equivalent (SWE) in the harsh environments of high mountain regions is a challenge. Here, we explore the use of muon counts to infer SWE. We deployed a muonic cosmic ray snow gauge ($\mu$-CRSG) on a Swiss glacier during the snow rich winter season 2020/21 (almost 2000 mm w.e.). The $\mu$-CRSG measurements agree well with measurements by a neutronic cosmic ray snow gauge (n-CRSG) and they lie within the uncertainty of manual observations. We conclude that the $\mu$-CRSG is a highly promising method to monitor SWE in remote high mountain environments with several advantages over the n-CRSG.

## 1 Introduction

The snow water equivalent (SWE) of the seasonal snowpack is a key variable of the hydrological and climate system and highly relevant for hydrological, glaciological and meteorological studies, especially in high mountain regions. However, operational monitoring of SWE in high mountain regions still poses considerable technical and logistic challenges because of the harsh environmental conditions (wind, icing, etc.) and the remoteness of the measurement sites (e.g., Kinar and Pomeroy, 2015; Nitu et al., 2018). As a result, temporally continuous and accurate SWE measurements in high mountain regions are very scarce and/or associated with significant uncertainties.

Several investigated methods take advantage of naturally occurring cosmic radiation to infer SWE temporally continuously. These methods make use of gamma radiation (e.g., Osterhuber et al., 1998; Choquette et al., 2008) or of neutrons from secondary cascades of cosmic rays (Kodama et al., 1975). The neutronic cosmic ray snow gauge (n-CRSG), a method proposed by Kodama et al. (1975), measures the attenuation of incoming secondary neutrons on the ground below the snowpack to infer SWE. This has proved successful (e.g., Wada et al., 1977; Kodama et al., 1979; Kodama, 1980; Avdyushin et al., 1982), especially for remote and harsh environments (e.g., Howat et al., 2018; Gugerli et al., 2019). Nonetheless, some drawbacks such as the limited measurement precision that can be achieved with a reasonably sized sensor have been identified (e.g., Gugerli et al., 2019).

A similar cosmic ray method measures neutrons scattered near the land-atmosphere boundary with a n-CRSG above the snowpack (e.g., Desilets et al., 2010; Rasmussen et al., 2012). The great advantage is that it is non invasive and offers a large footprint. However, it is limited to SWE amounts of around 600 mm w.e. in non-glacierized areas (Schattan et al., 2017).

Instead of using secondary neutrons as outlined above, we here investigate a muonic cosmic ray snow gauge ($\mu$-CRSG) to obtain temporally continuous SWE measurements. Cosmic ray muons are highly penetrating particles and thus not as sensitive to SWE as neutrons. But, the highly penetrating nature of muons also makes them far more abundant than neutrons at ground level, and provides a compensating statistical advantage over neutrons that should result in a better measurement precision. However, unlike neutrons, muons are unstable and can decay in mid flight. For this reason, muon intensity at ground level is

influenced by the distance traveled, or, more specifically, by the thickness of the atmosphere on any given day (e.g., Riádigos et al., 2020). Hence, there are several known and probably also unknown trade offs between neutrons and muons with consequences on inferring SWE from these measurements.

The aim of this study is to explore the use of muons to infer temporally continuous SWE in a high mountain glacierized site, and to provide a first-cut calibration function for the $\mu$-CRSG. The $\mu$-CRSG measurements are compared to manually obtained

SWE, to hourly SWE measurements obtained by a n-CRSG and to hourly snow depth measurements. Furthermore, we discuss the advantages of a $\mu$-CRSG with a focus on SWE monitoring in remote and harsh high mountain environments.

## 2 Study site and data

### 2.1 Study site

In December 2020, we deployed two $\mu$-CRSG (prototype *Bruno* provided by Hydroinnova LLC) on the Glacier de la Plaine

Morte in the Swiss Alps. The glacier has an area of 7.1 km$^2$ (2019) and elevation bands from 2650 m a.s.l. to 2800 m a.s.l. It is the largest plateau glacier of the European Alps (GLAMOS, 2020).

An automatic weather station with a n-CRSG (*SnowFox*$^{\text{TM}}$ provided by Hydroinnova LLC) was deployed in autumn 2016 on Plaine Morte (46°22.8′ N, 7°29.7′ E, 2689 m a.s.l., see Gugerli et al., 2019, for more information). The two $\mu$-CRSG were added to this station, one buried below the snow, i.e. lying on the glacier ice surface close to the n-CRSG, and one added at

the top of the station at 4.8 m height above the glacier ice surface (see Fig. S1). While only one n-CRSG is deployed, two $\mu$-CRSG are necessary to account for atmospheric influences on the muon count rates. For the n-CRSG, parameterizations have previously been applied and discussed to correct for changes in atmospheric pressure and incoming cosmic ray fluxes (e.g., Howat et al., 2018; Gugerli et al., 2019).

### 2.2 Data

This study encompasses four types of observational data sets. First, five manual SWE measurements were obtained between 16 December 2020 and 20 May 2021 by means of snowpits and snow cores, which complement a series of totally 22 manual SWE measurements between 20 Oct 2016 and 20 May 2021 at the same site (Gugerli, 2020). The uncertainty of these manual

observations is defined as the standard deviation of several observations during the same field day. Typically, the observations are taken within the same snow pit due to time restrictions. These snow pits and snow cores are located within a 30m radius of the station's mast. For each field campaign another location was used to avoid sampling a disturbed snowpack. SWE is calculated by multiplying the manually-obtained average bulk snow density with the autonomous and undisturbed daily snow depth observations by the ultra sonic ranger installed at the station. Second, hourly SWE obtained by a n-CRSG are available from 20 Oct 2016 to 13 August 2021 and validated with the 22 manual SWE measurements. Third, two $\mu$-CRSG were deployed on 16 December 2020 and provided hourly measurements until 13 August 2021. Fourth, hourly snow depth measurements from 16 December 2020 to 13 August 2021 obtained by a ultra sonic ranger mounted at a height of 4.8 m above the glacier surface are included for a further independent comparison.

From 16 December 2020 to 13 August 2021, 241 days of hourly neutron counts and 213 days of hourly muon counts were obtained. The data gaps within the muon count rates are due to unusual amounts of snow in the winter season 2020/21, which buried the solar panels and interrupted power supply. Since the solar panels of the n-CRSG are mounted higher up, and the measurement setup contains larger batteries, these measurements were not interrupted by the large snow amounts. However, the large amounts of snow also led to data gaps within the time series of snow depth measurements. These gaps occurred from 14 March 2021 to 22 March 2021 and from 5 May 2021 to 4 June 2021. The snow depth gaps were filled with measurements from a nearby high-altitude station, which are calibrated to our site (see supplement for further information).

## 3  Methods

To assess the performance of the $\mu$-CRSG, we (i) process neutron and muon counts to make them directly comparable and (ii) compare SWE inferred from neutron and muon counts over time. While the n-CRSG is an established method and conversion functions have been thoroughly assessed, using $\mu$-CRSGs is a novel approach and no data or conversion functions exist to the best of the author's knowledge. Hence, we derive a conversion function based on our manually obtained SWE observations.

### 3.1  Neutronic cosmic ray snow gauge (n-CRSG)

#### 3.1.1  Correcting neutron counts

The hourly neutron counts of the n-CRSG are first corrected for influences from incoming neutron fluxes and for variations in barometric air pressure. Following today's standard correction functions for sub-snow n-CRSG (e.g., Howat et al., 2018; Gugerli et al., 2019), we use hourly in situ pressure measurements ($p_i$ in hPa), the attenuation length at the site ($L$=132 hPa) and hourly neutron count rates ($F_{\mathrm{inc},i}$ in cps) from a reference neutron monitor located on Jungfraujoch in Switzerland (JUNG, http://www.nmdb.eu/) with a site-specific adjustment factor for Plaine Morte ($\beta$=0.95). The corrected hourly neutron counts ($N_{\mathrm{corr},i}$ in cph) are obtained as

$$N_{\mathrm{corr},i} = N_{\mathrm{raw},i} \cdot (\beta \cdot (\frac{F_{\mathrm{inc},i}}{F_{\mathrm{inc},0}} - 1) + 1) \cdot exp\left(\frac{p_i - p_0}{L}\right). \tag{1}$$

The reference values for $F_{\text{inc},0}$ (cph) and $p_0$ (hPa) correspond to the 24h mean from 12 July 2017 08 UTC to 13 July 2017 08 UTC.

### 3.1.2 Inferring SWE from neutron counts

The corrected neutron counts ($N_{\text{corr},i}$) are converted to hourly SWE ($SWE_{\text{n},i}$ in cm w.e.) by

$$SWE_{\text{n},i} = -\frac{1}{\Lambda_i} \cdot \ln \frac{N_{\text{corr},i}}{N_0} \tag{2}$$

where the variable $\Lambda_i$ is the effective attenuation length given by

$$\Lambda_i = \frac{1}{\Lambda_{\text{max}}} + \left( \frac{1}{\Lambda_{\text{min}}} - \frac{1}{\Lambda_{\text{max}}} \right) \cdot \left( 1 + exp \left( -\frac{\frac{N_{\text{corr},i}}{N_0} - a_1}{a_2} \right) \right)^{-a_3} \tag{3}$$

The snow free count rate ($N_0$ in cph) corresponds to the median of the corrected neutron counts ($N_0$=4146 cph) during the same 24h reference period used for the correction factors (12 July 2017 08 UTC to 13 July 2017 08 UTC). The unitless calibration factors $a_1$, $a_2$, $a_3$ are 0.31, 0.08 and 1.12, respectively. The attenuation lengths $\Lambda_{\text{max}}$ and $\Lambda_{\text{min}}$ are 114.4 cm and 14.1 cm, respectively (Howat et al., 2018; Gugerli et al., 2019).

To increase our confidence in the n-CRSG observations, we extend the previous validation of the n-CRSG on Plaine Morte from nine (Gugerli et al., 2019) to 22 manually obtained SWE estimates by snow pits and snow cores (Fig. 1). The 22 manual measurements are significantly and highly correlated with a coefficient of determination of 0.969 (Fig. 1a). On average, the n-CRSG agrees with the manually obtained SWE with an underestimation of -2% and an uncertainty of $\pm 10\%$ (one standard deviation, Fig. 1b). The root mean square error amounts to 112 mm w.e. Please also note that 50% of the manual field observations are obtained from snowpacks that are deeper than 1130 mm w.e.

### 3.2 Muonic cosmic ray snow gauge ($\mu$-CRSG)

We use two $\mu$-CRSG deployed on the glacier site; one below and one above the snowpack. Monitoring the incoming muon counts with the sensor above the snowpack allows to directly correct for the temporal variability caused by atmospheric effects such as air pressure variations and variations in incoming cosmic ray fluxes. Besides these effects, the muon intensity also depends on the temperature profile of the atmosphere. The temperature influences the production rate of muons (positive temperature effect) as well as its decay rate (negative temperature effect, e.g., de Mendonça et al., 2016; Riádigos et al., 2020).

### 3.2.1 Correcting muon counts

To obtain a time series of muon count rates corrected for atmospheric influences, we multiply the count rate under snow free conditions with the relative muon count rate ($f_{\mu,i}$). The relative count rate is derived as

$$f_{\mu,i} = \frac{\mu_{\text{sub},i}}{\mu_{\text{top},i}} \tag{4}$$

where $\mu_{\text{top},i}$ ($\mu_{\text{sub},i}$) is the hourly count rate of the $\mu$-CRSG above (below) the snowpack. We assume that atmospheric influences are manifested in the measurements of both devices and that the relative count rate only represents changes related to the

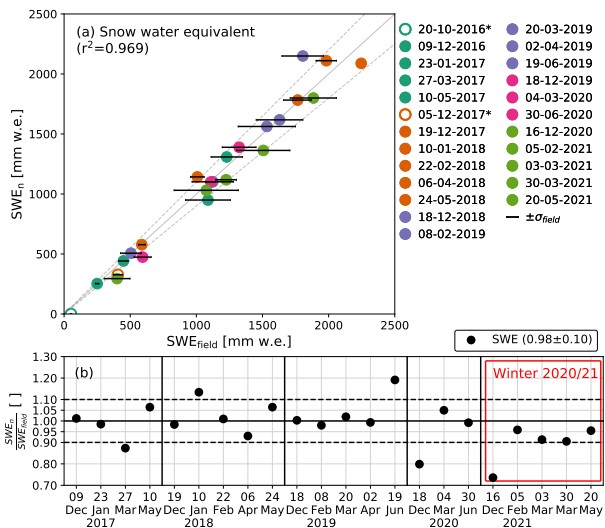

**Figure 1.** Validation of the n-CRSG. Panel (a) shows SWE derived by the n-CRSG compared to SWE derived manually with a $r^2$ of 0.969. The field observations from 20 Oct 2016 and 5 Dec 2017 were done while deploying the devices and are thus not taken into account for the validation. [Figure adapted from Gugerli (2020)]

snowpack (accumulation and ablation). With the relative count rate, we can derive the corrected muon count rate ($\mu_{\mathrm{corr},i}$ in cph) as

$$\mu_{\mathrm{corr},i} = \mu_{\mathrm{sub},0} \cdot f_{\mu,i}. \tag{5}$$

The variable $\mu_{\mathrm{sub},0}$ corresponds to the mean daily count rate under snow free conditions on the 12 August 2021 from 0 UTC until 23 UTC ($\mu_{\mathrm{sub},0}$=42202 cph). This measurement is obtained from the $\mu$-CRSG lying on the glacier ice surface to assure a direct comparison to the n-CRSG, which is also lying on the ice surface.

### 3.2.2 Inferring SWE from muon counts

The conversion function used to infer SWE from muon counts is derived by using the manual field observations on Plaine
Morte. Independent data obtained by descending the same prototype of the $\mu$-CRSG into Cochiti lake in New Mexico, USA (1702 m a.s.l., see supplement) exist. However, we cannot use these data directly because of the different locations, and more importantly, the different elevations. Nonetheless, the decreasing muon count rate with water depth suggests a discontinuous function with a transition in slope (muon attenuation lenght) between 1000 mm and 1500 mm water depth (see Fig. S2), and we base our assumption of a two-part conversion function on these measurements.

Our conversion function is derived by splitting the five available manual field measurements into two parts. For the first part of the discontinuous function, we use the observations from 16 Dec 2020 and 5 Feb 2021, where we fit an exponential function. The second part of the conversion function is obtained through a fit between the manual observations on 5 Feb 2021 and 20

Mai 2021. The discontinuous function transitions at the SWE amount obtained on 5 Feb 2021, which corresponds to a relative muon count rate of 0.65. This yields

$$
\begin{aligned}
&if \ \frac{\mu_{\text{sub},i}}{\mu_{\text{top},i}} <= 0.65 \quad SWE_{\mu,i} = -2646 \cdot \ln(\frac{\mu_{\text{sub},i}}{\mu_{\text{top},i}}) - 69 \\
&if \ \frac{\mu_{\text{sub},i}}{\mu_{\text{top},i}} > 0.65 \quad SWE_{\mu,i} = -5384 \cdot \ln(\frac{\mu_{\text{sub},i}}{\mu_{\text{top},i}}) - 1243
\end{aligned}
\tag{6}
$$

to convert relative muon counts to hourly SWE ($SWE_{\mu,i}$ in mm w.e.).

## 4 Results and discussion

### 4.1 Comparison of sub-snow neutron and muon counts

The evolution of neutron and muon counts over the winter season 2020/21 is presented in Fig. 2. When the muon detectors were deployed in December 2020, the snowpack had a depth of 140 cm with a SWE of 393±98 mm w.e. (16 December 2020). Hence, the muon counts above and below the snowpack differ in the beginning of the measurements (Fig. 2b). This difference increases with the deepening snowpack until beginning of June 2021. In June, snow ablation dominates and the difference between the sub and top $\mu$-CRSG decreases until they have similar count rates in August 2021, when the site becomes snow free.

The temporal variability of the corrected muon counts (Fig. 2b) correlate well with the temporal variability in corrected neutron counts (Fig. 2a). As with the n-CRSG, periods with snow accumulation show decreasing counts (e.g., mid January 2021 to beginning of February 2021) and periods with snow ablation increasing counts (e.g., mid June to end of July). In between the count rate remains stable (e.g., end December to mid January). Considering the uncorrected neutron and muon counts, temporal fluctuations related to atmospheric effects are very similar, too (Fig. 2).

Comparing neutron and muon counts, the counting statistics are highly different, which influences the uncertainty of these counts. The uncertainty of the count rate is defined as the square root of the count rate divided by the count rate itself. Gugerli et al. (2019) demonstrate that the main contributor to a low measurement precision of the n-CRSG, especially for deep snowpacks, is the uncertainty within the neutron counts. This precision is estimated through error propagation of a non-linear equation considering all variables (see Eq. 1) with their uncertainties (see Table 5 in Gugerli et al., 2019). The corrected neutron counts range between 379 cph and 4256 cph, and the corrected muon counts have count rates between 22540 cph and 42479 cph. The higher count rate strongly reduces the uncertainty of these measurements. While the uncertainty of the counts range between 1.5% and 5.1% for the neutrons, they lie between 0.5% and 0.7% for the muons. Note that these uncertainties only refer to the count rates and do not include potential systematic biases or influences by the parameterization of correction functions.

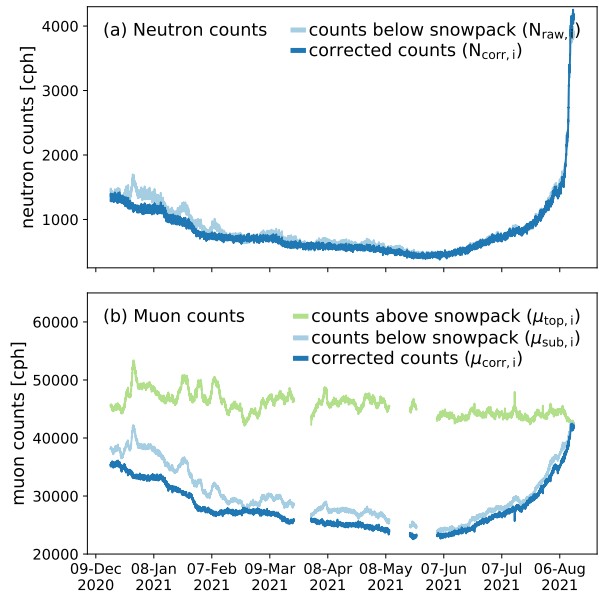

**Figure 2.** Neutron and muon counts from 16 Dec 2020 to 13 August 2021. Panel (a) shows neutron counts from the n-CRSG and (b) muon counts from the $\mu$-CRSG during the same time period. Please note the different scales for the y-axis.

## 4.2 Evaluation of SWE inferred by muon counts

The good agreement in the evolution of neutron and muon count rates presented in Fig. 2 shows the potential for using muon counts to infer SWE. Figure 3a shows an exponential relationship between the muon count rates and SWE at a daily resolution (manually obtained SWE) as well as at an hourly resolution (SWE obtained by the n-CRSG).

As suggested by the independent data obtained in a lake (cf. Sect. 3.2.2 and the supplement), a discontinuity within the relation between relative muon counts and SWE is manifested in Fig. 3a with the manually obtained SWE and SWE derived from the n-CRSG. Figure 3a indicates a potential transition between 750 mm w.e. and 1250 mm w.e., which is in line with the data obtained from lake experiments. Due to the snow rich winter season of 2020/21, only one field measurements (1065 mm w.e., 5 Feb 2021) is available within this transition bin. SWE amounts larger than 1065 mm w.e. (5 Feb 2021) are better represented. In the conversion function presented here we account for this transition in the attenuation length of muons with increasing SWE. To the best of the authors knowledge, however, no other data is currently available to derive a conversion function that is suitable for this glacierized site. Thus, our conversion function relies on the manual field measurements. While this results in a good agreement between $\mu$-CRSG SWE and n-CRSG SWE, some limitations remain. With the fit between relative muon count rates and manually obtained SWE, the condition of having 0 mm w.e. for a relative muon count rate of 1.0 is not fulfilled. Either a third part of the conversion needs to be introduced, or the fit needs to be repeated with more manual measurements. The transition within the conversion function could be caused by a softer component of the ionizing radiation from secondary

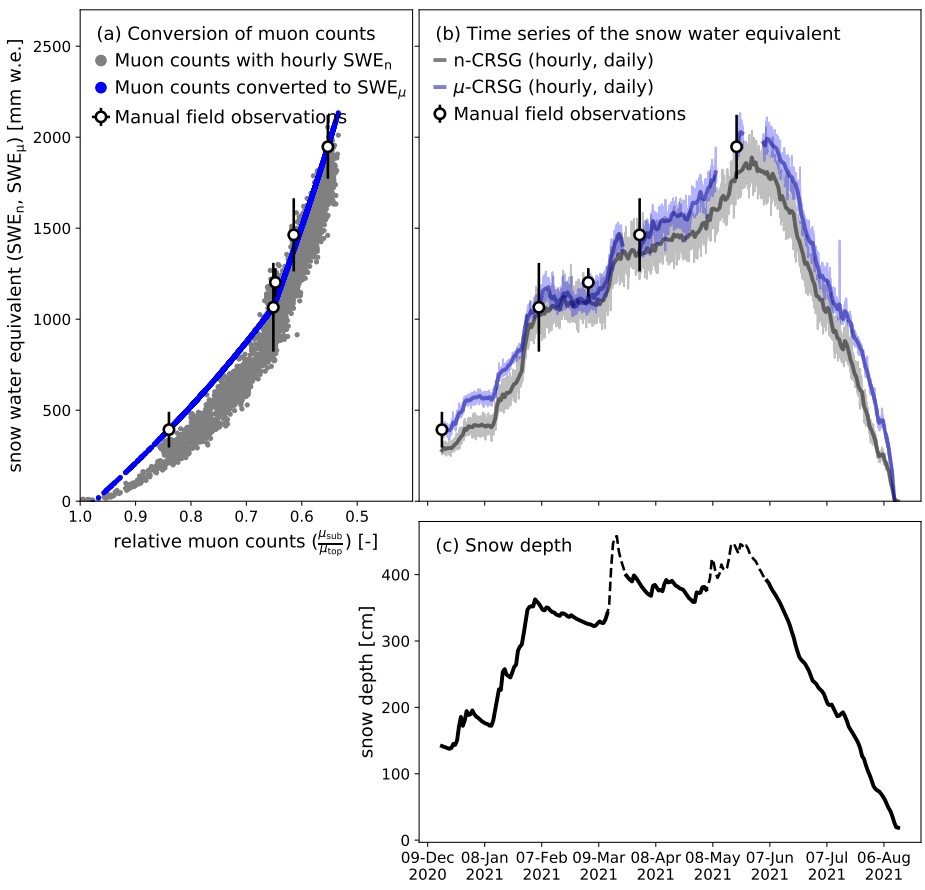

**Figure 3.** From muon count rates to SWE. Panel (a) shows the relative muon count rates plotted against SWE that is measured by the n-CRSG (grey dots) and measured manually (white dots). Blue dots represent the SWE that is directly inferred from the relative muon count rate with the conversion function given in Eq. 6. Panel (b) shows the time series of SWE inferred from neutron counts (grey) and muon counts (blue) at a daily resolution. Light grey and light blue represent hourly observations of the n-CRSG and $\mu$-CRSG, respectively. Panel (c) shows snow depth measured at the site (solid line) with data gaps that were complemented from a nearby high-altitude station (dashed line).

cosmic rays. The ratio of the soft and hard component could also be location and especially elevation dependent. Hence, a site-specific calibration could be necessary. Nonetheless, this remains highly speculative and further measurement experiments would be needed to investigate it in more depth. A robust statistical evaluation of the presented conversion function is not possible nor representative because only two manual field measurements remain independent.

Manually obtained SWE observations also carry uncertainties, which are depicted in Fig.3. These correspond to the standard deviation of several measurements obtained close to the sensor on the same day. Especially measurements within deep snowpacks are laborious and may have significant uncertainties due to limited access to deeper layers. The uncertainty of these field observations vary between 7% (3 March 2021) and 25% (16 Dec 2020). The latter was especially challenging to measure because of an unusually deep layer of light and powdery snow.

Moreover, the validation with manual field observations may include a spatial and methodological uncertainty. While there is a spatial distance between the n-CRSG, $\mu$-CRSG and the manual snow measurements, which are obtained at a different location on each field day, the site is rather flat and independent snow depth measurements have shown little variability within a 30 m radius around the station (Gugerli, 2020). The manual snow observations are based on two main approaches; short tube (55 cm long) samplings within snow pits and snow core samplings. To avoid an influence of the measurement approach, the estimated

bulk snow density is based on the average over several samples. Hence, we limit uncertainties related to the measurement approaches, and integrate them within the standard deviation over all samples obtained at the same day.

Furthermore, the deployment of the $\mu$-CRSG on 16 December 2020 disturbed the snowpack above the sub-snow $\mu$-CSRG. This disturbance does not seem to have a significant influence on the estimated SWE as the evolution in SWE agree well at the beginning of the season. In addition, the bias between the two devices could also be related to the strong underestimation of

the first manual field measurements (16 Dec 2020) by the n-CRSG.

Despite these limitations, we derive temporally continuous SWE from muon counts (Fig. 3b) that agree well with independent SWE measurements by the n-CRSG. Mostly, daily n-CSRG and $\mu$-CRSG lie within the uncertainty of the manual field observations (Fig. 3b). Note that the agreement with some of the manually-obtained field data is related to how the conversion function was derived.

Generally, the hourly SWE observations by the $\mu$-CRSG have less variability throughout the day than hourly SWE observations by the n-CRSG as their spread around daily averages is lower (Fig. 3b). Nonetheless, daily $\mu$-CRSG SWE show larger changes between days compared to n-CRSG SWE for SWE larger 1000 mm w.e. The inter-daily fluctuations in February 2021 could be related to two major Sahara dust events (5-6 Feb 2021 and 22-25 Feb 2021 MeteoSchweiz, 2021), but this remains speculative. More quantitatively, daily changes in SWE, i.e. the difference between the daily mean SWE of day 1 and day 0,

show a correlation of 0.64 between the n-CRSG and $\mu$-CRSG (Figure S3a). Daily decreases of SWE before the onset of snow melt (beginning of June 2021) may be related to snow erosion. However, an analysis with daily maximum wind speeds did not show conclusive results (see supplement). Noise within the muon count measurements, the two-part conversion function, and/or production and decay processes within the snowpack that may affect the top $\mu$-CRSG differently than the sub-snow one may influence daily changes within the SWE observations by the $\mu$-CRSG. Further investigations including simulations

of cosmic-ray muon production and decay are required to analyze and quantify these influences. Such investigations, however,

are beyond the scope of this work.

Independent snow depth measurements provide a further comparison to the n-CRSG and $\mu$-CRSG estimates. Accumulation periods identified by the n-CRSG and $\mu$-CRSG agree well with increases in snow depth measurements (Fig. 3b and c). However, periods with decreases in snow depth cannot be directly compared to SWE estimates. Compaction of the snowpack, for example, would result in a decreasing snow depth but not in a decreasing SWE value.

### 4.3   The potential of the muonic cosmic ray snow gauge to monitor SWE in high mountain regions

Our results from two $\mu$-CRSG deployed on a glacierized sites confirm the promising approach of using $\mu$-CRSG to infer temporally continuous SWE on glacierized high mountain sites. With the improved counting statistics, the uncertainty of the count numbers is reduced by almost a factor of 10 compared to neutron count uncertainties. As Gugerli et al. (2019) show, the uncertainty in the count numbers are the largest contributor to the overall uncertainty of these measurements. Based on the theoretical precision estimation, the $\mu$-CRSG promises to infer sub-daily SWE estimates with a higher precision than the n-CRSG. In addition, the hourly observations vary less around the daily mean for the $\mu$-CRSG than for the n-CRSG (Fig. 3b). Nonetheless, the $\mu$-CRSG contain some inter-daily fluctuations that are larger in the $\mu$-CRSG estimates than the n-CRSG. To understand these, further investigations are needed.

The $\mu$-CRSG has additional important advantages regarding its suitability and applicability in remote high mountain environments compared to a n-CRSG. The $\mu$-CRSG is technically more robust and lighter, it consumes less energy and is overall cheaper in its production as it does not require exotic fill gases or elaborate cleaning procedures during manufacture.

### 5   Conclusions and perspectives

This study presents the potential of monitoring SWE in glacierized high mountain environments by means of muon counts. We infer SWE from a relative muon count rate from two $\mu$-CRSGs deployed on an alpine glacier. The direct comparison to independent n-CRSG observations demonstrates the proof of concept of inferring SWE from muons and highlights the great potential for glacierized high mountain regions. This study further advances our knowledge and possibilities of monitoring SWE accurately and reliably in technical challenging environments.

The main limitation of our study is the number of manually obtained SWE observations. Due to logistical and financial restrictions no further manual measurements were possible, and to the best of the author's knowledge, no other data for further analysis for the glacierized site are available. This limitation is addressed by including hourly SWE measurements by a n-CRSG that have extensively been validated (cf. Sect. 3.1.2).

In future studies, more manual measurements, measurement experiments and simulations can improve our understanding of these devices. Further measurements can be used to test and validate the presented conversion function in more depth. Moreover, correction functions for incoming variations, which are similar to the parameterizations of the n-CRSG, can be derived. Potential correction functions for the temperature effect on the muon intensity have been previously investigated (e.g., Ganeva et al., 2013), and should be analysed for the application of a sub-snow $\mu$-CRSG. Once such influences can be accounted for,

only one $\mu$-CRSG deployed below the snowpack would be needed. Being cheaper and lighter than the n-CRSG, more devices can be deployed covering larger areas and thus reducing uncertainties in area-wide SWE by remote sensing and/or modelling

approaches.

*Data availability.* All data will be available in a future repository.

*Author contributions.* RG conducted the analysis and prepared the manuscript with input from all authors. DD contributed to the data analysis and data interpretation. All authors contributed to the design of this study.

*Competing interests.* RG and NS declare that they have no competing interests. The author DD is the owner of Hydroinnova LLC.

*Disclaimer.* TEXT

*Acknowledgements.* This research is supported by the Swiss National Science Foundation (SNFS, grant no. 200021_178963). We are very grateful to all field helpers who joined us on Plaine Morte to obtain manual snow observations. We acknowledge the NMDB database (www.nmdb.eu) founded under the European Union's FP7 programme (contract no. 213 007), and the PIs of individual neutron monitors at: IGY Jungfraujoch and NM64 Jungfraujoch (Physikalisches Institut, University of Bern, Switzerland) for the data provided to correct the

neutronic cosmic ray snow gauge counts for the incoming cosmic ray flux. Last but not least, we thank the two anonymous reviewers and the editor for their constructive feedback and suggestions that significantly improved the paper.

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
