# Peer review of "Brief communication: Application of a muonic cosmic ray snow gauge to monitor the snow water equivalent on alpine glaciers"

_The Cryosphere, 2021_

## Author Comment (AC1)

**Referee #1 – Authors reply**

*In the following, we provide a point-to-point answer to the referee's comments.* The referee's comments start with RC:*, authors replies begin with AR: and are formatted italic with light-blue color.*

A markup of manuscript changes are shown in boxes with new text in dark blue and
* * *
RC: The authors present a prototype sensor to measure SWE based on muonic cosmic ray. They derive SWE by fitting the count rate with a few manual measurements which were performed during one season on a Swiss glacier. Reliable methods to measure SWE temporally continuously in alpine environments are urgently needed. Studies as presented in this paper are therefore highly welcome and fit well into TC. I liked reading the manuscript, which has a clear structure and illustrative figures. The language is, with a few exceptions, easy to read. I suggest to accept the manuscript as soon as the following points have been addressed.

*AR: We thank the referee for his/her dedicated time to our manuscript and the constructive feedback, which will significantly improve our manuscript.*

RC: The described potential for reliable daily data seems much too optimistic when looking at Fig. 3a. The daily fluctuation shown therein make physically no sense. Even neglecting the large fluctuation in February, the daily signal (for the µ-CRSG and n-CRSG) often demonstrates strong negative changes during the accumulation season (e.g. April), which make only sense if you have large snow erosion. Please show a comparison with the daily snow depth change and discuss this problem. In light of the problem with daily values, a sentence like "the µ-CRSG promises to infer sub-daily SWE estimates with a higher precision than the n-CRSG" is quite bold!

*AR: We added snow depth to Fig. 3, where accumulation periods agree fairly well with increases in both sensor's measurements. However, ablation periods cannot be compared directly to snow depth measurements because of the process of densification. While the snowpack densifies, SWE remains constant. Nonetheless, the additional comparison to snow depth further builds trust in our measurements.*

*Our conclusion that the µ-CRSG provides a higher precision than the n-CRSG is based on a theoretical approach. Following a Poisson distribution, the uncertainty within the count rates corresponds to the square root of the count rate divided by the total count rate. With lower count rates, this precision is lower. Since the muons are so abundant on the Earth's surface, the count rate is significantly higher, and thus, the theoretical precision too. We adapted the sentence to emphasize our reasoning behind it. To better understand the daily fluctuations, however, more measurements and different field experiments would be necessary.*

[Figure]

Fig. 3 From muon count rates to SWE. Panel (a) shows the relative muon count rates plotted against SWE that is measured by the n-CRSG (grey dots) and measured manually (white dots). Blue dots represent the SWE that is directly inferred from the relative muon count rate with the conversion function given in Eq.6. Panel (b) shows the time series of SWE inferred from neutron counts (grey) and muon counts (blue) at a daily resolution. Light grey and light blue represent hourly observations of the n-CRSG and μ-CRSG, respectively. Panel (c) shows snow depth measured at the site (solid line) with data gaps that were complemented from a nearby high-altitude station (dashed line).

[Section 4.2]

The independent snow depth measurements provide a further comparison to the n-CRSG and μ-CRSG estimates. Accumulation periods identified by the n-CRSG and μ-CRSG agree well with increases in snow depth measurements (Fig.3b and c). However, periods with snow erosion cannot be directly compared to the snow depth measurements because a decrease in snow depth can have two causes; densification of the snowpack, snow erosion or ablation. Therefore, a comparison of such time periods is not straightforward.

[Section 4.3]

Based on the theoretical precision estimation, the μ-CRSG promises to infer sub-daily SWE estimates with a higher precision than the n-CRSG. In addition, the hourly observations vary less around the daily mean for the μ-CRSG than for the n-CRSG (Fig.3b). Nonetheless, the μ-CRSG contain some inter-daily fluctuations that are larger in the μ-CRSG estimates than the n-CRSG. To understand these, further investigations are needed.

RC: You defined the uncertainty of the manual observations as the standard deviation of several observations during the same field day. These measurements usually have different SWE because neither the glacier surface nor the snow depth is perfectly homogenous. The account for these differences all same day measurements are usually referred to common snow depth (assuming that

the bulk snow density is constant), which is often the one from the snow depth sensor, because this location should be undisturbed throughout the whole season. The same should be done when comparing the CRSG measurements with the manual measurements as both measurements usually have a different snow depth. Could please elaborate your procedure to deal with this issue?

*AR: Thank you for this comment. We agree that it is an important point. In most field campaigns, especially those where we had a deep snowpack, all measurements are taken within the same snow pit. Often, we did not have enough time nor human power to excavate several deep snow pits on the same day. Snow cores are drilled where the snow pit is excavated afterwards. Hence, the spatial variability between the different measurements should be reduced to a minimum.*

*The spatial variability between the manual and automatic measurements is not straightforward to evaluate. The n-CRSG is placed about 7m away from the ultra sonic ranger to reduce influences of the metallic mast parts on the n-CRS (see newly added figure Fig. S1). In turn, we reduce the influence of the n-CRSG on the ultra sonic ranger footprint.*

[Figure]

Fig. S1: Measurement setup on the Glacier de la Plaine Morte with the sub-snow neutronic cosmic ray snow gauge and the top and sub muonic cosmic ray snow gauge. [Photo courtesy: M. Huss]

*Gugerli (2020) provides an example to demonstrate the low spatial variability at the site on Plaine Morte. For example, on 4 March 2020, 18 random snow depth measurements within 30m radius of the station were obtained. On average, the snow depth is 377±11 cm, which shows a spatial variability of ±3%. Simultaneously, the average snow depth differs by 3% from the ultra sonic measurement (389cm).*

*While the spatial variability of snow depth is low, the standard deviation of the SWE observations varies between 4% and 25% for measurements between October 2016 and May 2021. On the day with a 25% variation between the SWE observations (16 Dec 2020), the observations are based on three snow core samples. In conclusion, the variability in snow depth is low compared to the variability in samples within the same snow pit and/or over several snow pits.*

*Following the comment of referee #1 and #2, we now base our validation of the n-CRSG on re-calculated field data. Instead of calculating SWE for each sample, we derive an average manually-*

*obtained bulk snow density and multiply it with the undisturbed and autonomous snow depth observation by the ultra sonic ranger. The validation of the n-CRSG does not change significantly with this new approach. Generally, the temporal variability in the agreement between the field observations and the n-CRSG estimates improves.*

*We modified the manuscript as follows.*
* * *
[2.2. Data]

The uncertainty of these manual observations is defined as the standard deviation of several observations during the same field day. Typically, the observations are taken within the same snow pit due to time restrictions. These snow pits and snow cores are located within a 30m radius of the station's mast. For each field campaign another location was used to avoid sampling a disturbed snowpack. SWE is calculated by multiplying the manually-obtained average bulk snow density with the autonomous and undisturbed daily snow depth observations by an ultra sonic ranger.

[3.1.2. Inferring SWE from neutron counts]

The 22 manual measurements are significantly and highly correlated with a coefficient of determination of 0.969 (Fig. 1a). On average, then-CRSG agrees with the manually obtained SWE with an underestimation of -2% and an uncertainty of ±10% (one standard deviation, Fig. 1b). The root mean square error amounts to 112mm w.e. Please also note that 50% of the manual field observations are obtained from snowpacks that are deeper than 1130 mm w.e.

[4.2. Evaluation of SWE inferred by muon counts]

Moreover, the validation with manual field observations may include a spatial and methodological uncertainty. While there is a spatial distance between the n-CRSG, μ-CRSG and the manual snow measurements, which are obtained at a different location on each field day, the site is rather flat and independent snow depth measurements have shown little variability within a 30m radius around the station (Gugerli, 2020). The manual snow observations are based on two main approaches; short tube samplings within snow pits and snow core samplings. To avoid an influence of the measurement approach, the estimated bulk snow density is based on the average over several samples. Hence, we limit uncertainties related to the measurement approaches, and integrate them within the standard deviation over all samples obtained at the same day.
* * *
RC: I cannot get the SWE numbers shown in Fig 3a with the given Equation 6 using the relative muon count rate as input? For a muon count rate of 0.7 for example, I get 89 and not ca. 1000 mm SWE as given by Figure 3a?

*AR: Thank you for checking our calculations! We agree that it is crucial to have them correct. Indeed, there is a conversion mistake of a factor of 10 (from cm to mm). We apologize for this mistake and corrected the equation so that we get mm w.e. and not cm w.e.*

*We now calculate a SWE of 875 mm w.e. with a differential muon count rate of 0.70 (Eq. 6). This is also the value presented in Fig.3a.*

*We also adapted the figure in the supplement, where the fit corresponds to cm water depth and not to mm water depth.*

RC: Is there a physical explanation behind the two-part conversion function, i.e. why there is a change in slope between 1000 and 1500 mm water depth in Fig. S1?

*AR: This is a very interesting and important point, which requires more research. We are convinced that we also measure a softer component in addition to what is fairly certainly a pure muon component at depth. To distinguish between these two components, however, further field deployments and experiments would be necessary.*

*We added the following to the manuscript.*
* * *
[Section 4.2]

With the fit between relative muon count rates and manually obtained SWE, the condition of having 0mm w.e. for a relative muon count rate of 1.0 is not fulfilled. Either a third part of the conversion needs to be introduced, or the fit needs to be repeated with more manual measurements. The transition within the conversion function could be caused by a softer component of the ionizing radiation from secondary cosmic rays. The ratio of the soft and hard component could also be location and especially elevation dependent. Hence, a site-specific calibration could be necessary. Nonetheless, this remains highly speculative and further measurement experiments would be needed to investigate it in more depth. A robust statistical evaluation of the presented conversion function is not possible nor representative because only two manual field measurements remain independent.
* * *
RC: Looking at Fig. 3a it may rather be a three-part conversion function?

*AR: Yes, as mentioned in the manuscript, it may even be a three-part function. A function of the type used for the n-CRSG (e.g., Howat et al., 2018; Gugerli et al., 2019), which was derived for a similar purpose, might also be appropriate here.*

RC: Please better discuss if the application of a μ-CRSG in other locations will always need some site-specific calibration or not.

*AR: We believe a site specific calibration function could be needed given that the ratio of the two components (softer and harder component) could be location dependent. We believe this ratio should particularly depend on elevation. It should be straightforward to answer this question definitively through shielding experiments at different elevations. Another approach would be to try to eliminate or minimize one of the components, leaving a simpler and more universal calibration function.*
* * *
[Section 4.2]

The transition within the conversion function could be caused by a softer component of the ionizing radiation from secondary cosmic rays. The ratio of the soft and hard component could also be location and especially elevation dependent. Hence, a site-specific calibration could be necessary. Nonetheless, this remains highly speculative and further measurement experiments would be needed to investigate it in more depth. A robust statistical evaluation of the presented conversion function is not possible nor representative because only two manual field measurements remain independent.

[Section 5]

In future studies, more manual measurements, further measurement experiments and simulations can improve our understanding of this measurement approach in addition to validate the presented conversion function.
* * *
Minor points:

L17: … measures the attenuation of incoming secondary neutrons on the ground below the snowpack to infer SWE.

*AR: Done*

L21: with an n-CRSG above the snowpack

*AR: Done*

L43: Would be nice to have a picture to demonstrate the setup of the instruments.

*AR: We fully agree. We did not include a photo because of the figure limitation given for the manuscript type "Brief communication" in The Cryosphere. We added a photo of the setup in the supplement as Fig. S1 (see above) instead.*

L45: …parameterizations have previously been investigated (please reference).

*AR: We changed the sentence as follows:*

For the n-CRSG, parameterizations have previously been  applied and discussed to correct for changes in atmospheric pressure and incoming cosmic ray fluxes (e.g., Howat et al. 2018; Gugerli et al. 2019).

---

## Author Comment (AC2)

**Referee #2 – Authors reply**

*In the following, we provide a point-to-point answer to the referee's comments.* The referee's comments start with RC:*, authors replies begin with AR: and are formatted italic with light-blue color.*

A markup of manuscript changes are shown in boxes with new text in dark blue and

RC: This brief communication on the ‚Application of a muonic cosmic ray snow gauge to monitor the snow water equivalent on alpine glaciers' by Gugerli et al. gives a comprehensive introduction to a new cosmic ray sensor to monitor SWE at a point scale. The manuscript is well structured and gives comprehensive information on the applied methods and results. Moreover, the authors try to discuss their results carefully and point out potential uncertainties and further steps. Of course, it would be nice to get more detailed information on the method itself, however, for the chosen type of manuscript (brief communication) the length and amount of given information is well suited. In general, I see a great need in investigation of new sensor systems as presented here, as we still lack continuous in situ SWE measurements in alpine areas for various applications. I only have some minor points:

*AR: We thank the Referee #2 for the time dedicated to our manuscript and the positive and constructive feedback, which will significantly improve it.*

RC: Was the glacier surface already covered by snow at the date of installation (6 December 2020)? If yes, the natural snow cover was most likely destroyed by shovelling, which could have an impact on the results, especially at the beginning of the winter period. Please state on this.

*AR: Unfortunately, we were not able to deploy the muon detectors before the snowpack started building due to logistical constraints. At the time of deployment of the muon detectors, the snowpack was already 140 cm deep (snow depth). In addition, the snow was very powdery and manual measurements were challenging.*

*The referee is right that it could influence the results of the µ-CRSG. As the n-CRSG was already deployed prior to the first snowfall, this instrument is not influenced. Comparing the inferred SWE from the n-CRSG with the µ-CRSG at the beginning of the season, the evolution is very similar. Due to the nature of the derivation of the conversion function, which uses the first manual measurement that is strongly underestimated by the n-CRSG, a bias is to be expected. We added a paragraph discussing this point.*

[Section 4.1]

When the muon detectors were deployed in December 2020, the snowpack had a depth of 140 cm with a SWE of 393±98 mm w.e. (16 December 2020).

[Section 4.2]

Furthermore, the deployment of the µ-CSRG on 16 December 2020 disturbed the snowpack above the sub-snow µ-CSRG. This disturbance does not seem to have a significant influence on the estimated SWE as the evolution in SWE agree well at the beginning of the season. In addition, the

> bias between the two devices could also be related to the strong underestimation of the first manual field measurements (16 Dec 2020) by the n-CRSG

RC: What is the distance [m] between the buried neutron comic ray sensor and the buried muonic cosmic ray sensor? At what distances where the manual measurements carried out? I agree with Reviewer 1 on his point 2 – referring the measurements to a common snow depth should be applied if available (if not, please discuss this issue carefully).

*AR: Concerning the distances between the n-CSRG and µ-CRSG, we added a photo in the supplement. The manuscript type "Brief communication" is restricted to three figures (see responses to Referee #1)*

*Concerning the validation of the SWE estimates by the n-CRSG, we adapted the validation of the n-CRSG as also suggested by Referee #1. The adapted validation is based on SWE derived by the manually obtained bulk snow density multiplied with the autonomous snow depth estimates. It generally improves our results.*

RC: Please add the RMSE (besides R²) to describe the accuracy between manual SWE measurements and n-CRSG-derived SWE (Section 3.1.2).

*AR: We added the RMSE value (see manuscript excerpt above).*

> The 22 manual measurements are significantly and highly correlated with a coefficient of determination of 0.969 (Fig. 1a). On average, then-CRSG agrees with the manually obtained SWE with an underestimation of  2% and an uncertainty of  ±10% (one standard deviation, Fig. 1b). The root mean square error amounts to 112 mm w.e. Please note that 50% of the manual field observations are obtained in snowpacks deeper than 1130 mm w.e.

RC: The two-part or eventually also three-part conversation function needs more explanation and background information (Section 3.2.2).

*AR: In general, we agree. But we would also like to point out that this is a brief communication of a study with limited funding. We wish to show that results are promising enough to justify continued research on the method, and that the remaining questions are tractable.*

*We added the following to the manuscript to provide more explanations and background information.*

> [Section 4.2]
>
> In the conversion function presented here we account for this transition in the attenuation length of muons with increasing SWE. To the best of the authors knowledge, however, no other data is currently available to derive a conversion function that is suitable for this glacierized site. Thus, our conversion function relies on the manual field measurements. While this results in a good agreement between µ-CRSG SWE and n-CRSG SWE, some limitations remain. With the fit between relative muon count rates and manually obtained SWE, the condition of having 0mm w.e. for a relative muon count rate of 1.0 is not fulfilled. Either a third part of the conversion needs to be introduced, or the fit needs to be repeated with more manual measurements. The transition within the conversion function could be caused by a softer component of the ionizing radiation from secondary cosmic rays. The ratio of the soft and hard component could also be location and especially elevation dependent. Hence, a site-specific calibration could be necessary. Nonetheless, this remains highly speculative and further measurement experiments would be needed to investigate it in more depth.

A robust statistical evaluation of the presented conversion function is not possible nor representative because only two manual field measurements remain independent.

[Section 5]

In future studies, more manual measurements, further measurement experiments and simulations can improve our understanding of this measurement approach in addition to validate the presented conversion function.

RC: (How) does the footprint of this in situ measurements change with an increasing snowpack?

*AR: The extent of the footprint of the n-CRSG remains subject to further studies, and thus only assumptions can be made. We believe that the footprint is cylindrical, and depends on the depth of the snowpack. Currently, a paper by colleagues, which investigates the footprint of the sub-snow n-CRSG with a modelling approach, is in preparation. However, we cannot refer to it yet.*

RC: In addition, could you give an assumption how the accuracy of the novel method changes with an increase of SWE?

*Concerning the accuracy of the µ-CRSG, we assume a similar behavior as for the n-CRSG, i.e., the precision is mainly defined by the statistical uncertainty of the count rate, which depends on the depth of the snowpack. Gugerli et al. (2019) estimate the precision of the n-CRSG with error propagation and an uncertainty assumption for each parameter used to calculate SWE from the neutron count rate. As we deployed two µ-CRSG, the same approach would only include the uncertainty of the count rates by the sensors propagated through the conversion equation.*

*We modified the following paragraph.*

[Section 4.3]

Based on the theoretical precision estimation, the µ-CRSG promises to infer sub-daily SWE estimates with a higher precision than the n-CRSG. In addition, the hourly observations vary less around the daily mean for the µ-CRSG than for the n-CRSG (Fig. 3b). Nonetheless, the µ-CRSG contain some inter-daily fluctuations that are larger in the µ-CRSG estimates than the n-CRSG. To understand these, further investigations are needed.

---

## Author Response (AR2)

Dear Editor,

We sincerely thank the two referees for reviewing our manuscript and the positive evaluation of the revised manuscript.

We have considered the major and minor points of Referee #1 and added some further information on the daily fluctuations of the measurements in the manuscript and the supplement.

Please find a detailed point-by-point answer to the referee's comments below.

Yours sincerely, Rebecca Gugerli on behalf of all authors

**Referee #1 – Authors reply**

*In the following, we provide a point-to-point answer to the referee's comments.* The referee's comments start with RC:, *authors replies begin with AR: and are formatted italic with light-blue color.*

A markup of manuscript changes are shown in boxes with new text in dark blue and <del>removed text in red.</del>

RC: Thanks to authors for the clear and concise reply. The paper definitely improved. However, I still miss one important point, which I believe needs to be discussed, before acceptance.

**AR:* We thank the Referee #1 for the time dedicated to our revised manuscript, and the positive and constructive feedback to our changes.**

RC: My original remark "Even neglecting the large fluctuation in February, the daily signal (for the  $\mu$ -CRSG and n-CRSG) often demonstrates strong negative changes during the accumulation season (e.g. April)" has not been addressed properly. You now just assume that the frequent decreases in SWE are caused by snow erosion without any further analysis. I would at least expect that you check if periods with clearly decreasing SWE (Fig. 3b) correlate with high wind speed and concurrent decreasing snow depth. If not, the decreasing SWE values must be caused by something else. It might be related to the two-part conversion function, because the unexpected SWE decreases only happen above 1000 mm SWE!

**AR: Thank you for this comment. We apologize for not having been clear about our assumptions of the SWE decreases. There are certainly many effects that may explain these inter-daily fluctuations.**

In general, the change in SWE from a day to another (inter-daily fluctuation) can be related to changes of the snowpack as well as influences by the cosmic ray fluxes and/or the conversion function. Environmental conditions, which cause snow drift, deposition and/or sublimation, may lead to increasing or decreasing amounts of SWE. In addition, the statistical variability in the count rates becomes more important with increasing SWE.

We included some further analysis for the inter-daily fluctuations and added the results to the manuscript in Section 4.1. An additional analysis with daily maximum wind speeds was added to the supplement in a new section and referenced in the manuscript.

**[Section 4.1.]**

Generally, the hourly SWE observations by the µ-CRSG has have less variability throughout the day than hourly SWE observations by the n-CRSG as their spread around daily averages is lower (Fig. 3b). Nonetheless, daily µ-CRSG SWE fluctuates more in February compared to the show larger changes between days compared to n-CRSG SWE Two for SWE larger 1000 mm w.e. The inter-daily fluctuations in February 2021 could be related to two major Sahara dust events (5-6 Feb 2021 and 22-25 Feb 2021, MeteoSchweiz, 2021)could be related to these fluctuations, but this remains speculative. Apart from this period, the temporal fluctuations are consistent between More quantitatively, daily changes in SWE, i.e. the difference between the daily mean SWE of day 1 and day 0, show a correlation of 0.64 between the n-CRSG and µ-CRSG SWE(Figure S3a). Daily decreases of SWE before the onset of snow melt (beginning of June 2021) may be related to snow erosion. However, an analysis with daily maximum wind speeds did not show conclusive results (see supplement). Noise within the muon count measurements, the two-part conversion function, and/or production and decay processes within the snowpack that may affect the top  $\mu$ -CRSG differently than the sub-snow one may influence daily changes within the SWE observations by the  $\mu$ -CRSG. Further investigations including simulations of cosmic-ray muon production and decay are required to analyze and quantify these influences. Such investigations, however, are beyond the scope of this work.

[Supplement: Section 4 Daily fluctuations analyzed with maximum daily wind speed]

Hourly wind speed is measured at the station on the Glacier de la Plaine Morte from 20 October 2016 to 13 August 2021. The n-CRSG measurements in a deep snowpack are limited in their temporal resolution due to the counting rate statistics at this site. Hence, the following analysis is based on daily measurements.

Inter-daily fluctuations of n-CRSG and  $\mu$ -CRSG correspond to the difference of the daily mean SWE from day 1 to day 0. If the change is positive (negative), an increase (decrease) in SWE is observed. Figure S3a shows the correlation between daily changes of n-CRSG and  $\mu$ -CRSG.

Daily maximum wind speed is derived from hourly measurements of maximum wind speed. From 16 December 2020 to 13 August 2021, 238 days of daily maximum wind speed observations are available. On average, the daily maximum wind speed is 10.0±3.9 ms-1

Figure S3b shows a statistical summary of these daily differences as a function of maximum daily wind speed bins. If the daily maximum wind speed is lower 10 ms-1, no change or a minimal decrease in SWE is observed by both CRSG's. For wind speeds higher 14 ms-1, both devices show more days with increasing SWE amounts. Almost 80% of days with maximum wind speeds higher 14 ms-1 occur from December 2020 to April 2021. Please note that snow drift is not only a function of wind speed, but also of the snow density of the top layer of the snowpack. With our measurement setup, however, we have no means of deriving a reliable snow density of the affected snow layer. With snow depth, only a bulk snow density can be derived.